# Preparation of Nanocellulose-Based Aerogel and Its Research Progress in Wastewater Treatment

**DOI:** 10.3390/molecules28083541

**Published:** 2023-04-17

**Authors:** Jiaxin Zhao, Xushuo Yuan, Xiaoxiao Wu, Li Liu, Haiyang Guo, Kaimeng Xu, Lianpeng Zhang, Guanben Du

**Affiliations:** 1Yunnan Provincial Key Laboratory of Wood Adhesives and Glued Products, Southwest Forestry University, Kunming 650224, China; 2Jiaxing Key Laboratory of Molecular Recognition and Sensing, College of Biological, Chemical Sciences and Engineering, Jiaxing University, Jiaxing 314001, China

**Keywords:** aerogel, nanocellulose, adsorption, wastewater treatment, application

## Abstract

Nowadays, the fast expansion of the economy and industry results in a considerable volume of wastewater being released, severely affecting water quality and the environment. It has a significant influence on the biological environment, both terrestrial and aquatic plant and animal life, and human health. Therefore, wastewater treatment is a global issue of great concern. Nanocellulose’s hydrophilicity, easy surface modification, rich functional groups, and biocompatibility make it a candidate material for the preparation of aerogels. The third generation of aerogel is a nanocellulose-based aerogel. It has unique advantages such as a high specific surface area, a three-dimensional structure, is biodegradable, has a low density, has high porosity, and is renewable. It has the opportunity to replace traditional adsorbents (activated carbon, activated zeolite, etc.). This paper reviews the fabrication of nanocellulose-based aerogels. The preparation process is divided into four main steps: the preparation of nanocellulose, gelation of nanocellulose, solvent replacement of nanocellulose wet gel, and drying of nanocellulose wet aerogel. Furthermore, the research progress of the application of nanocellulose-based aerogels in the adsorption of dyes, heavy metal ions, antibiotics, organic solvents, and oil-water separation is reviewed. Finally, the development prospects and future challenges of nanocellulose-based aerogels are discussed.

## 1. Introduction

Among the most fundamental requirements for survival, water is absolutely crucial for all forms of life on Earth. However, the rapid growth of the world population, the construction of modern cities, and the development of industrialization have led to the pollution of a large amount of water quality environment. This issue has been an ongoing global problem. Therefore, water environment management has become one of the most critical parts of global environmental governance [1,2]. Due to various industrial and human activities, there is a large amount of harmful substances in wastewater. It mainly includes dyes, heavy metal ions, petroleum, organic solvents, and antibiotics. The long-term accumulation of these harmful substances in nature can cause serious complications for humans and animals and cause serious damage to the environment [3,4]. Therefore, it is necessary to eliminate these harmful substances before discharging wastewater to reduce their contaminating effects on earthly organisms and the environment. Reverse osmosis, ion exchange, chemical precipitation, filtration, solvent extraction, oxidation, coagulation, and adsorption are common methods for removing hazardous substances from wastewater [5,6]. In these technologies, the main principle of adsorption is the interaction between the contaminant and the adsorbent, which causes the contaminant to be immobilized on the surface of the adsorbent by the interaction. They primarily comprise electrostatic interactions, hydrogen bonding, ion exchange, van der Waals forces, and hydrophobic interactions [7]. In addition, adsorption technology has many advantages, such as eco-friendliness, low processing costs, and the availability of large quantities of materials for use as adsorbents [3]. Therefore, adsorption technology is considered to be one of the simplest and easiest to operate and implement [8]. Due to these characteristics, adsorption technology is used extensively for wastewater remediation.

In wastewater treatment, various adsorbents are made in large quantities. For example, activated carbon, graphene, nanocellulose, metal organic skeletons, etc. In addition to the above-mentioned adsorbents, aerogels are also considered excellent candidates for adsorbents in water purification. It is an ultra-lightweight, highly porous solid material. It has a three-dimensional structure, a low density, a large specific surface area, a high porosity, and excellent vibration damping properties [9,10,11]. Kistler [12] created the first aerogel in the early 1930s by supercritically drying the wet gel to eliminate the liquid. In the following decades, different kinds of aerogels were made by researchers, such as inorganic aerogels [13], polymer-based aerogels [14], and nanocellulose-based aerogels [15]. Compared with organic aerogels, inorganic aerogels have obvious disadvantages, such as weak mechanical strength and fragility [15]. Nanocellulose-based aerogel has become the leading environmental material with the advantages of being green, renewable, biodegradable, etc. It is widely used in the fields of adsorbent material, catalyst carrier, filter material, heat insulation material, etc. [16,17]. Therefore, nanocellulose-based aerogels have received a lot of attention from researchers in various countries.

This paper first introduces the preparation of nanocellulose as well as nanocellulose-based aerogel. Figure 1 shows a simple process for the preparation of nanocellulose-based aerogels. Then, the paper reviewed the recent research progress of nanocellulose aerogel in various applications such as dyes, heavy metal ions, antibiotics, organic solvents, and oil-water separation. Lastly, recommendations for future research, which explore its challenges and shortcomings and provide strategies for future study. To further extend the application of nanocellulose-based aerogel in wastewater environment treatment.

## 2. Preparation of a Nanocellulose-Based Aerogel

Nanocellulose-based aerogel is the latest generation of aerogel made of nanocellulose. It not only retains the same qualities as traditional aerogel, but it also has the attributes of being renewable, green, non-toxic, low-cost, and biodegradable nanocellulose [18]. Usually, nanocellulose aerogels are prepared in four steps: preparation of the nanocellulose, gelation of the nanocellulose, solvent replacement of the nanocellulose wet gel, and drying of the nanocellulose wet gel. The structure and characteristics of nanocellulose-based aerogels are affected by each preparation process. The microstructure and characteristics of nanocellulose aerogels are also affected by different cellulose sources and production methods.

### 2.1. Preparation of Nanocellulose

Cellulose is the most prevalent and extensively dispersed natural polymer in nature. It is constituted of D-glucopyranose units connected by 1,4 glycosidic linkages [19,20]. Nanocellulose is a natural cellulose with a one-dimensional nanoscale size. There are three types of nanocellulose: cellulose nanocrystals (CNC), cellulose nanofilaments (CNF), and bacterial cellulose (BC) [21,22,23]. Table 1 shows the classification of nanocellulose. They differ in size, morphology, and preparation methods [24]. Nanocellulose offers several benefits, including a large specific surface area, biodegradability, renewable energy, and a high aspect ratio [25,26,27]. Nanocellulose is an important substrate in the production of nanocellulose-based aerogels. The different sources of its preparation methods and the quality of its preparation will directly affect the performance and application of nanocellulose-based aerogels.

The common preparation methods of nanocellulose are chemical, mechanical, chemical-mechanical, and bio-enzymatic [31]. Table 2 describes the classification of nanocellulose preparation methods. The chemical method commonly used is 2,2,6,6-tetramethylpiperidin-1-yloxy (TEMPO) oxidation of cellulose. Xue et al. [32] prepared CNF by oxidizing cellulose pulp at pH = 10 using the TEMPO/NaBr/NaClO system. CNF has a high specific surface area and a consistent and concentrated pore size distribution of 10–15 nm. Chen et al. [33] prepared nanocellulose from bamboo, cotton lint, and sisal using TEMPO, NaBr, and NaClO solutions with crystallinities of 60.1%, 66.5%, and 83.9%, respectively. The widths of all three samples were in the range of 5–14 nm and showed well-dispersed forms. Wen et al. [34] prepared highly thermally stable and hydrophobic lignocellulosic nanoprimer fibers from poplar high-yield pulp by controlling different amounts of NaClO using the TEMPO oxidation system. Sun et al. [35] utilized the TEMPO oxidation system to effectively manufacture cellulose nanocrystals from bleached wood pulp. The average length is 200.7 nm, the average diameter is 5.8 nm, and the aspect ratio is 34.4. However, the TEMPO oxidation technique necessitates more stringent pH control of the reaction fluid [36]. In addition, the TEMPO oxidation procedure may result in the existence of trace levels of radical species in the sample, which may restrict the material’s utility [37].

Mechanical methods mainly use high cutting forces to break down the fiber, such as the ball milling method, the high pressure homogenization method, the ultrasonic method, etc. [45]. Although the mechanical method is less polluting to the environment, it consumes more energy, and the uniformity of the prepared nanocellulose (CNF) is poor [46]. Nevertheless, using chemical pretreatment followed by mechanical treatment can minimize mechanical treatment energy consumption while improving CNF quality. Common methods of chemical pretreatment include amine functionalization [47], carboxymethylation [48], phosphorylation [49], acetylation [50], nitration [51], etc. Chemical pretreatment creates new functional groups on the fiber surface, further broadening the application of CNF. Henschen et al. [52] prepared oxalic acid cellulose using paper pulp and oxalic acid dihydrate. Nanocellulose was then produced by high-pressure homogenization. The length and breadth of the produced nanocellulose averaged 350 nm and 3–4 nm, respectively. The bio-enzymatic method is an enzymatic treatment of fibers, and enzymatic treatment is considered a green and sustainable method for CNF preparation. Zhang et al. [53] generated nanocellulose by bio-enzymatic-assisted ultrasonication after pretreating poplar wood with the steam blasting method. The widths were in the range of 20–50 nm. They featured high aspect ratios and network entanglement structures. Tao et al. [54] used xylanase to pretreat sugarcane bagasse pulp and combined it with a mechanical method to prepare cellulose nanogenic fibers (CNF). The research revealed that the thermal stability of CNF produced after enzyme treatment increased with crystallinity. Nie et al. [55] prepared CNF suspensions with higher crystallinity by pretreating unbleached eucalyptus pulp with xylanase combined with high-pressure homogenization. It has higher dispersibility and rheology.

Acid hydrolysis is a typical process for preparing CNC. That is, the acid is utilized to break down the amorphous area of cellulose while retaining the crystalline region. This method can prepare CNC with high crystallinity, etc. The acid hydrolysis preparation process has been very well developed and industrialized into production, but the acid hydrolysis method requires high equipment materials. Kassab et al. [56] first extracted purified cellulose microfibers (CMF) from Juncus plant stems by chemical methods. Sulfated cellulose nanocrystals (S-CNC) were then synthesized using the acid hydrolysis of sulfuric acid and a citric/hydrochloric acid combination. The diameter of the S-CNC is 7.3 ± 2.2 nm, the length is 431 ± 94 nm, and the crystallinity is 81%. Rashaad et al. [57] used acid hydrolysis of bamboo fibers and isolated needle-like CNC with 86.96% crystallinity from the bamboo fibers. The CNC yield rate is around 22%. BC is a naturally occurring nanostructured polymeric substance that is mostly manufactured by bacteria. Other polymers like lignin, hemicellulose, and pectin are not present in BC [58,59]. The performance of BC is determined by the medium composition and the type of bacteria utilized [45]. Salari et al. [60] obtained xyloglucan bacilli in beet molasses, cheese whey, and standard Hestrin-Schramm (HS) medium, thereby producing BC. They prepared bacterial cellulose nanocrystals (BCNC) from the resulting BC by hydrolyzing it with sulfuric acid. The diameter and length of BCNC were 25 ± 5 nm and 306 ± 112 nm, respectively. Xu et al. [61] used sweet potato residue (SPR) hydrolysate as a medium, and the SPR hydrolysate was used directly with little to no inhibitors, which facilitated the subsequent synthesis of SPR-BC. SPR-BC has a crystallinity of 87.39% and a maximum thermal degradation temperature of 263 °C. Superior to the synthetic media preparation of BC.

### 2.2. Gelation of Nanocellulose

Nanocellulose gelation is a vital process for the creation of nanocellulose-based aerogels. Nanocellulose forms three-dimensional network structures during gel formation, and these network structures increase the strength of the gel. Physical cross-linking and chemical cross-linking are the two mechanisms by which gelation is formed [62]. Physical cross-linking is typically caused by inter- or intramolecular hydrogen bonding as well as physical inter-molecular entanglement. To develop a network structure, chemical cross-linking typically requires the addition of additional cross-linking agents. In general, physical cross-linking produces a less stable structure than chemical cross-linking [63]. The physical gelation system of nanocellulose is primarily dependent on the creation of intermolecular as well as intramolecular hydrogen bonds with the hydroxyl groups present on the surface of nanocellulose. Usually, the prepared nanocellulose is uniformly dispersed in water and then spontaneously forms into a hydrogel under hydrogen bonding [64]. Xing et al. [65] obtained nanocellulose from durian peel. At a concentration greater than 0.5%, it may spontaneously cross-link at room temperature to produce a hydrogel. The experimental results show that CNF aerogels containing 1% CNF exhibit a sheet-like structure supported by fibers. CNF aerogel has the highest compression ability and is 99% pore size. Shaheed et al. [66] isolated cellulose from peanut shells and then formed the gel spontaneously by high-intensity ultrasound. Pure nanocellulose aerogels (NFC aerogels) were synthesized by solvent substitution and freeze-drying. Then Cu-BTC/NFC aerogels were prepared by adding a metal-organic backbone (Cu-BTC) to the nanocellulose gels through the direct mixing method. The procedure for preparation is depicted in Figure 1. Heath et al. [67] dispersed nanocellulose in water using an aqueous-phase dispersion method, allowing spontaneous gel formation. Then, the nanocellulose-based aerogel with a density of 78 mg/cm^3^ and a specific surface area of 605 m^2^/g was prepared by the solvent replacement technique and the supercritical CO_2_ drying method. The chemical cross-linking of nanocellulose gelation happens mostly during the sol-gel technique’s preparation. Zhu et al. [68] synthesized two distinct aerogels from CNCs and CNFs using the sol-gel technique. According to the findings, at 85% strain, CNCs and CNFs aerogels had compressive strengths of 269.5 kPa and 299.5 kPa, respectively. Gupta et al. [69] prepared aerogels based on nano-protofibrillated cellulose and polymethylsesquioxane using the sol-gel method. A sol-gel preparation of aerogels can increase their mechanical characteristics and thermal stability and have a unique 3D network of interconnected nanoscale particles [70]. Unfortunately, the sol-gel process for preparing aerogels is time-consuming. So the rate of gelation is accelerated by adding chemical cross-linking agents to the liquid sol or by changing the physical conditions (temperature, pH, ultrasonic treatment, etc.) [71]. Mu et al. [72] designed and synthesized polyorganosiloxanes containing polyorganosiloxanes by free radical polymerization. The polyorganosiloxanes could be covalently cross-linked with cellulose nanofibers (CNF). The resulting aerogels had a 3D structure with a high specific surface area of 53.88 m^2^/g. Ruan et al. [73] used calcium chloride as a green cross-linking agent for cellulose nanocrystals (CNC). Then freeze-dry the CNC gels. A CNC aerogel with good stability and ultra-light performance (density of 0.036 g/cm^3^) was successfully prepared.

### 2.3. Solvent Substitution of Nanocellulose Wet Gel

Solvent replacement is the process of replacing organic solvents with water in a wet gel system. It prevents the gel skeleton from collapsing during the drying process [74]. The selection of a suitable organic solvent is the most important part of the solvent replacement process. If the chosen organic solvent is destructive to the structure of the nanocellulose hydrogel itself, then the structure and characteristics of the nanocellulose aerogel will be affected. Generally, organic solvents with a lower surface tension than that of water, such as ethanol, *tert*-butanol, and acetone, are used. These organic solvents can effectively reduce the capillary pressure inside the pore and protect the pore structure of aerogel [75]. In addition, the selection of a suitable organic solvent needs to be combined with the drying method. Aerogels prepared by the freeze-drying method are usually selected with *tert*-butyl alcohol. Ethanol and acetone are frequently used in the preparation of supercritical CO_2_-drying aerogels. Aerogel prepared by the atmospheric pressure drying method uses acetone, which is normally selected. Wu et al. [76] prepared biosynthesis, *tert*-butyl alcohol solvent replacement, and atom transfer radical polymerization to make functionalized biomass-derived bacterial cellulose aerogels with nano-network structure and high porosity. Zhang et al. [77] prepared CNC aerogels using different concentrations of *tert*-butanol solvent substitution and freeze-drying. Li et al. [75] prepared cellulose nanofibril-based aerogels using the acetone solvent replacement approach and atmospheric pressure drying. The results showed that the specific surface area of the prepared aerogel was 22.4 m^2^/g and the density was 58.82 mg/cm^3^. Ciftci et al. [78] prepared CNF-based hydrogels from cellulose nanofibers (CNFs) by ultrasonication, using ethanol as a replacement solvent. Then CNF aerogels were produced by drying with supercritical CO_2_. The aerogel was discovered to have a low density (0.009–0.05 g/cm^3^), high porosity (99%), and a high surface area (72–115 m^2^/g).

### 2.4. Drying of Nanocellulose Wet Gel

Drying is the final and most crucial stage in the production of nanocellulose-based aerogels. Owing to the high number of micropores present in wet gels. When utilizing conventional drying techniques, the bending of the gas-liquid interface could cause capillary pressure, which could cause the gel pores to collapse and break. However, the supercritical CO_2_ drying method and the freeze-drying method can prevent this phenomenon [63]. Thus, these two drying methods are routinely utilized to make nanocellulose-based aerogels. According to the different ways of freezing, the freeze-drying method can be divided into liquid nitrogen freeze drying and refrigerator freeze drying. Li et al. [79] successfully prepared anisotropic nanofiber aerogels loaded with modified UIO-66-EDTA by liquid nitrogen-directed freeze-drying technique. The liquid nitrogen-directed freeze-drying process is shown in Figure 2. Wu et al. [80] prepared aerogels with porous honeycomb structures from nanocellulose sols of poplar wool fibers (PCF). The density of this aerogel is only 0.3–0.4 mg/cm^3^, the porosity is greater than 99%, and the difference between the pyrolysis temperatures is very small. Qiu et al. [81] prepared nanocellulose aerogels using different solvents and different drying techniques. The results show that the aerogels prepared by ionic liquid and freeze-drying have the structure of a fibrous network with pores that are about 200 nm in size. The aerogels made from a NaOH/urea solution and supercritical drying also have a 3D fibrous network structure, but their pore size is smaller, approximately 50 nm. Wang et al. [82] used calcium chloride solution to prepare nanocellulose aerogels by spontaneous gelation with *tert*-butanol solvent replacement and freeze-drying techniques. It has a shrinkage of 5.89%, a specific surface area of 164.9666 m^2^/g, and an average pore size of 10.01 nm. Yang et al. [83] obtained chemically cross-linked cellulose nanocrystals (CNCs) aerogels using a supercritical CO_2_ drying method. CNCs aerogel has a density of just 5.6 mg/cm^3^, high porosity (99.6%), and a complete spatial network structure. Wang et al. [84] utilized supercritical CO_2_ drying to create the nanocellulose-based aerogel. It had a high specific surface area of 353 m^2/^g, the average pore size was 8.86 nm, and the shrinkage was 4.03%. Despite the fact that the supercritical CO_2_ drying method can keep an aerogel’s three-dimensional structure from collapsing. However, it is costly and energy intensive, which limits its application. Currently, some studies have started to prepare nanocellulose-based aerogels by the atmospheric pressure drying method. Fu et al. [85] successfully prepared CNF-SiO_2_ composite aerogels by atmospheric pressure drying at a temperature of 80 °C. Li et al. [86] obtained ultralight and porous cellulose-based aerogels using an atmospheric pressure drying method. The results of the analysis indicate that the aerogel had a porosity of over 98%, a density of 18 mg/cm^3^, and a specific surface area of over 30 m^2^/g. Zhang et al. [87] used an atmospheric pressure drying method to make graphene/cellulose nanocrystal hybrid aerogels with variable mechanical strengths. Georg et al. [88] first modified the hydrophobicity by immersing triphenylmethyl into the suspension of microcrystalline cellulose. The hydrophobic cellulose aerogel was then dried at an atmospheric pressure of 80 °C. It has a good skeletal structure. The atmospheric pressure drying method is simple and low-cost, and it can dry nanocellulose-based aerogels in large quantities. However, the process of atmospheric pressure drying method is not yet mature.

## 3. Application of Nanocellulose-Based Aerogel in Wastewater Treatment

The 3D network structure of nanocellulose-based aerogels, porosity in the micro/mesoporous range, large specific surface area, surface rich in hydroxyl groups, and easy functionalization modification make them good adsorbents [89]. Table 3 briefly summarizes the properties of nanocellulose-based aerogels compared with those of other natural material-based aerogels. Presently, nanocellulose-based aerogels are effectively used to treat wastewater. Researchers have developed a number of nanocellulose-based aerogels. They can be used to adsorb dyes, heavy metal ions, antibiotics, organic solvents, and oil-water separation.

### 3.1. Adsorption of Dyes

Most of the dyes in wastewater come from paper, leather, textiles, etc. They are water-soluble polymeric organic compounds with color-forming groups [94]. Dyes have a major impact on aquatic organisms. They have a significant propensity to chelate metal ions, which is hazardous to fish and other creatures and inhibits aquatic plant development [95]. Therefore, there is a global need to solve the problem of dye pollution. Recently, the use of nanocellulose-based aerogels for dye adsorption has attracted a lot of attention. Shaheed et al. [66] prepared Cu-BTC/NFC aerogel composites by compounding a metal organic backbone (Cu-BTC) with nanocellulose (NFC) using the direct mixing method. The experiments revealed that it has a good ability for adsorbing Congo red (CR). The maximal capacity for absorption is 39 mg/g. Wang et al. [96] composed a new green composite aerogel (CGS) by ultrasonically dispersing graphene oxide-silica (GO-silica) in a cellulose nanofiber suspension (CNF) and then freeze-drying the mixture. The findings showed that the composite aerogel exhibited excellent adsorption ability for methylene blue (MB) with a maximum value of 608.4 mg/g compared to the pure CNF aerogel (max = 328.3 mg/g). Xie et al. [97] prepared a high mechanical strength sodium alginate/cellulose nanofiber/polyethyleneimine composite aerogel (SCP) by combining cellulose nanofiber, sodium alginate, and polyethyleneimine. According to the results, the maximum adsorption of CR and methyl orange (MO) by SCP composite aerogel at the optimal pH (pH = 2.0–5.0) was 2007.48 mg/g and 2253.38 mg/g, respectively. Yu et al. [98] successfully synthesized composite aerogels (CNF-GnP) by assembling cellulose nanofibers (CNF) and graphene nanoplates (GnPs). It was shown that the maximum adsorption amount of CNF-GnP aerogel for CR was 585.3 mg/g, and the maximum adsorption amount for MB was 1178.5 mg/g. The binary dye absorption capacity of CNF-GnP hybrid aerogels is superior to that of pure CNF or GnP. In addition, using ethanol as a desorption agent, over 80% of CR or MB may be eluted from CNF-GNP, demonstrating the reusability of these aerogels. Maatar et al. [99] prepared cellulose aerogels based on cationic cellulose nanofibers (Q-CNF) by chemically cross-linking the nanofibers with aliphatic triisocyanates. The surface was enriched with trimethylammonium chloride functional groups. The Q-CNF aerogel was proven to be an effective adsorbent for anionic dyes and to be resistant to water disintegration. The adsorption capacities of Q-CNF for red, blue, and orange dyes were 160 mg/g, 230 mg/g, and 560 mg/g, respectively. Huo et al. [100] cross-linked CNF hydrogels by calcium ions and then immersed them in dopamine solution. Polydopamine (PDA) was used to modify the surface of CNF to produce PDA@CNF composite aerogel (PCNF). The surface area of PCNF composite aerogels is large (368.15 m^2^/g), and the packing density is low (27.2 mg/cm^3^). The experiments showed that PCNF aerogel has a strong adsorption capacity for MB. With a starting dye level of 50 mg/L, the maximum amount of MB that could be adsorbed was 208 mg/g. Grishkewich et al. [101] used dimethyl ammonium chloride (DADMAC), N,N’-methylenebis(acrylamide) (MBAA), functionalized cellulose nanofibers (CNFs), and (3-mercaptopropyl) trimethoxysilane (MPTMS) via the thiol-ene click reaction to prepare compressible aerogels (DADMAC-MBAA modified CNF-silica aerogels). It was demonstrated that the aerogel’s maximal MO adsorption capacity was 186.7 mg/g.

In addition, the surface charge properties of aerogels have an effect on dye adsorption. Yi et al. [102] successfully synthesized aramid nanofibers/bacterial cellulose (ANFs/BC) aerogels by a physical gelation process. The aerogel can selectively adsorb cationic dyes in wastewater. Thanks to the hydrophilic and negative charge of the surface. The cationic dye MB was well adsorbed by aerogels, whereas the anionic MO was poorly adsorbed. Figure 3 depicts the adsorption mechanism of ANFs/BC aerogel on dye MB. The dye removal efficiency for MB was 98.8%, much higher compared to the pure BC aerogel (27.9%). Nia et al. [103] synthesized a new silica-cellulose aerogel by chemical cross-linking. The aerogel has a density of 0.107 g/mL, a porosity of approximately 93.0%, and a surface with positively and negatively charged functional groups. The adsorption capacity was 270 mg/g for MB and 300 mg/g for MO. Jiang et al. [104] prepared cellulose nanofibril-based aerogels from TEMPO-oxidized CNF using *tert*-butanol solvent replacement and the freeze-drying method. It is highly effective at removing cationic malachite green dye from water. It has a specific surface area of 193 m^2^/g and a maximum adsorption of 212.7 mg/g. This happens because the negatively charged carboxylic acid group on the surface attracts the positively charged malachite green (MG) dye. It makes it easier to take out cationic dyes. A new type of adsorption study is also emerging. Yang et al. [105] effectively fabricated a new CO_2_-responsive cellulose nanofibril aerogel. When the aerogel received CO_2_ stimulation, the aerogel had maximal adsorption capacities of 598.8 mg/g, 621.1 mg/g, and 892.9 mg/g for MB, naphthol green B (NGB), and MO, respectively. It also exhibits excellent recyclability, as it retains its adsorption characteristics even after 20 cycles. Figure 4 depicts the controlled and recyclable dye removal process of this aerogel induced by CO_2_.

Table 4 summarizes the performance of nanocellulose-based aerogels in dye removal and compares it with chitosan-based aerogels.

### 3.2. Adsorption of Heavy Metal Ions

Heavy metal ions are dangerous to people’s health because of their non-biodegradability and tendency to build up in water. Heavy metal ion pollution is often considered one of the most serious contaminants [110,111]. Chemical precipitation, physical adsorption, ion exchange, bioremediation, etc. are all ways to get rid of heavy metal ions [112,113,114]. Physical adsorption technology is widely used because of its environmental protection and simple preparation process. Therefore, researchers have developed a variety of adsorbent materials. Among them, nanocellulose-based aerogels are the most promising materials owing to their degradability, regenerability, and ease of modification. Lam et al. [115] found that the ability of nanocellulose-based aerogels to hold heavy metals depends mainly on their active sites and porous structure. If the nanocellulose-based aerogels are inherently less hydrophilic and have fewer active sites, they need to be modified. Geng et al. [116] used TEMPO oxidized nanogenic fibrillated cellulose (TO-NFC) and aminopropyltrimethoxysilanes (APTMs) as raw materials. They synthesized 3D macroscopic aminosilylated nanocellulose aerogels (APTMs modified TO-NFC). The research findings indicate that the adsorption capacities of APTMs modified TO-NFC aerogels for Cu^2+^, Cd^2+^, and Hg^2+^ were 99.0 mg/g, 124.5 mg/g, and 242.1 mg/g, respectively. Mo et al. [117] prepared three-dimensional layered porous cellulose nanofiber/polyacrylamide composite aerogels by a simple in situ physical/chemical double cross-linking. Figure 5 shows the adsorption mechanism of this composite aerogel and a picture of the cycle of adsorption and regeneration. It could absorb as much as 240 mg/g of Cu^2+^. Cu^2+^ removal efficiency was maintained at over 80% after 10 cycles of adsorption and regeneration. Wang et al. [118] prepared aerogels with shape memory (MOF@CA) by compounding nanocellulose (CNF) with polyvinyl alcohol (PVA) and a metal organic backbone (MOF). Studies have shown that MOF@CA has a low density (9.8–11.2 mg/cm^3^), high porosity (99.4–99.5%), and good elasticity in both air and water. MOF@CA had adsorption capabilities of 123 mg/g for Pb^2+^ and 70.53 mg/g for Cu^2+^, respectively.

Another study, Li et al. [119], made NFC/PEI hybrid aerogels for the adsorption of Cu^2+^ and Pb^2+^ by cross-linking nanofibrillated cellulose (NFC) and polyethyleneimine (PEI) through electrostatic bonding. The greatest levels of Cu^2+^ and Pb^2+^ adsorbed by this aerogel were 175.44 mg/g and 357.44 mg/g, respectively. Guo et al. [120] synthesized a novel polyethyleneimine (PEI)-grafted porous cellulose@PEI aerogel (CPA) by a glutaraldehyde cross-linking process between polyethyleneimine (PEI) amine groups and hydroxyl groups. It was demonstrated that CPA has a maximum adsorption capacity for Cr^6+^ of 229.1 mg/g. Hong et al. [121] prepared PEI@CNF aerogel by grafting polyethyleneimine (PEI) onto the scaffold of cellulose nanofibers (CNFs). Its adsorption capacity for Cu^2+^ was 135.1 mg/g. In the context of other metal ions, it shows a high selectivity for Cu^2+^. Several studies have shown that composite aerogels have a stronger adsorption capacity than pure nanocellulose aerogels and maintain structural integrity after multiple applications. Wei et al. [122] integrated ferric tetroxide (Fe_3_O_4_) nanoparticles and nanocellulose to prepare magnetic hybrid aerogels. The experiments showed that the mixed aerogel adsorption effectiveness for Cr (VI) ions was best. The adsorption process is depicted in Figure 6a. In addition, as shown in Figure 6b, the aerogel was experimentally proved to absorb Pb (II) and Cu (II) from a water solution. Lei et al. [123] synthesized a composite aerogel by combining cellulose with a metal organic backbone (UIO-66-NH_2_). It has an equilibrium adsorption capacity of 89.40 mg/g for Pb^2+^ and may be reused without noticeable performance loss for more than 5 cycles. Wang et al. [124] obtained cross-linked CNF aerogels by cross-linking CNFs with polyamide epichlorohydrin resin solution by freeze-drying and vacuum drying. During static uranium adsorption, it displayed rapid adsorption kinetics and a high adsorption capacity (440.60 mg/g). Li et al. [125] developed amine-functionalized cellulose-based aerogel beads (CGP) for effective simultaneous adsorption, reduction, and chelation of Cr (VI). As a result of its high electrostatic attraction to Cr (VI), CGP has a maximum capacity for adsorption of 386.40 mg/g at 25 °C. Shahnaz et al. [126] made composite aerogel (NCNB) by combining customized nanobentonite with nanocellulose/chitosan. Evaluation of its potential to absorb heavy metals from wastewater. It was shown that the aerogel could remove Cr^6+^, Co^3+^, and Cu^2+^ with a maximum adsorption efficiency of 98.90%, 97.45%, and 99.01%, respectively.

Table 5 summarizes the performance of nanocellulose-based aerogels in metal ion removal and compares it with chitosan-based aerogels.

### 3.3. Adsorption of Antibiotics

An antibiotic is an anti-microbial compound. Common antibiotics include macrolides, lincosamides, tetracyclines, phosphate esters, etc. [132]. Antibiotics are mainly sourced from the pharmaceutical industry, hospitals, animal farms, municipal garbage, and wastewater from wastewater treatment plants [133]. Antibiotics are heavily used because of their ability to act specifically on disease-causing bacteria and fungi by killing or inhibiting their growth in human or animal hosts. However, over-application has resulted in large amounts of antibiotics being exposed to the natural environment.

The most common organic micropollutant found in wastewater is antibiotics, posing a severe environmental threat [134]. The application of nanocellulose-based aerogels in antibiotic adsorption has been a research hotspot in recent years. Yao et al. [135] used a one-step ultrasonic technique to create a cellulose nanofiber/graphene oxide hybrid aerogel (CNF/GO). It was utilized for antibiotic adsorption in water. The aerogel achieved a more than 69% removal rate of antibiotics. The adsorption amounts for chloramphenicol, macrolides, quinolones, β-lactams, sulfonamides, and tetracyclines were 418.7 mg/g, 291.8 mg/g, 128.3 mg/g, 230.7 mg/g, 227.3 mg/g, and 454.6 mg/g, respectively. In addition, the aerogel can still be reused after ten cycles with no appreciable loss in adsorption ability. In a separate study, Wang et al. [136] used a one-step ultrasonic approach to create cellulose nanofibrils/graphite oxide hybrid (GO-CNF) aerogels. The aerogel was shown to remove 73.9%, 79.1%, and 81.5%, 79.5% of tetracycline (TC), chlortetracycline (CTC), doxycycline (DXC), and oxytetracycline (OTC), respectively. For TC, CTC, OTC, and DXC, the maximal theoretical adsorption capacities of GO-CNF were 343.8 mg/g, 396.5 mg/g, 386.5 mg/g, and 469.7 mg/g, respectively. In the same study, Wang et al. [137] incorporated cellulose nanofibers (CNFs) into graphene oxide nanosheets by dispersing them. The composite graphene oxide (GO)/cellulose nanogels were then prepared by combining freeze-drying techniques. The adsorption experiments demonstrated that the composite aerogel had a maximum adsorption capacity for TC of 47.3 mg/g. Experiments on adsorption could maintain the removal rate of TC at about 97% after three cycles. Wei et al. [138] will prepare bacterial cellulose (BC) aerogel by freeze-drying technique. Then it was put into a tube furnace for carbonization to finally obtain carbon-bacterial cellulose aerogel (BCCA). At a temperature of 298 K, the maximum adsorption capacities of BCCA for chloramphenicol (CAP), norfloxacin (NOR), and sulfamethoxazole (SMX) were 525 mg/g, 1926 mg/g, and 1264 mg/g, respectively. Liu et al. [139] first obtained modified cellulose (RCA) by chemical-physical double cross-linking with epichlorohydrin (ECH). The RCA was then coated with polyaniline (PANI) via in situ polymerization. Lastly, a metal-organic backbone (ZIF-67) was grown in situ on the PANI-coated modified cellulose to prepare a ZIF-67/PANI/RCA aerogel composite adsorbent. According to the analysis, ZIF-67/PANI/RCA aerogel had an adsorption capability of 409.55 mg/g for TC with good recovery. After six adsorption and desorption cycles, the removal efficiency of TC remained over 94%.

In another study like it, Cui et al. [140] uniformly loaded zeolite imidazole ester skeleton-8 (ZIF-8) onto the surface of cellulose aerogel (CCA) by in situ growth. Finally, it was made into ZIF-8 zeolite cellulose aerogel (ZCCA). It was shown that ZCCA exhibited excellent adsorption performance for enrofloxacin (ENR), besides having a maximal capacity for adsorption of 172.09 mg/g. Ruan et al. [141] first prepared a cellulose nanocrystals (CNCs)/silica (SiO_2_) mixture solution. The subsequent step was to add a certain mass fraction of polyvinyl alcohol (PVA) to the mixture of CNCs and SiO_2_. Finally, the freeze-drying approach was utilized to make a PVA-assisted CNC/SiO_2_ aerogel. The research revealed that ciprofloxacin (CIP) was taken up by PVA-assisted CNCs/SiO_2_ composite aerogels mostly through hydrogen bonding, π-π interactions, electrostatic interactions, and hydrophobic interactions. Figure 7 displays the PVA-assisted CNCs/SiO_2_ aerogel adsorption mechanism. The greatest adsorption capacity for CIP in the PVA-assisted CNCs/SiO_2_ aerogel was 163.34 mg/g.

Table 6 demonstrates the performance of nanocellulose-based aerogels for antibiotic removal and compares them with chitosan-based aerogels.

### 3.4. Oil-Water Separation and Adsorption of Organic Solvents

Oil pollution has a great impact on the water environment and ecosystems. In addition to oil pollutants, industrial oily wastewater also poses a severe hazard to ecosystems and human health [62]. Organic solvents in wastewater generally originate from the textile, printing leather, and chemical industries, as well as from the cleaning and polishing of furniture components. Such as benzonitrile, chloroform, glycol ether, methylene chloride, acetone, toluene, ethylbenzene, and xylene [145]. Organic solvents in water, like oil, can damage the marine environment and have a serious impact on the survival of aquatic and terrestrial plants and animals [146]. Oil pollution and organic solvents pose substantial harm to ecosystems, human life, economies, etc. Common methods used to clean up petroleum contamination and organic solvents include in-situ combustion, mechanical methods, chemical treatment, bioremediation, and adsorption [147]. The adsorption method is considered an economic and effective method to deal with oil contamination and organic solvents [145]. Because it costs less to make, uses less energy, and does not cause secondary contamination. The advantages of naturally renewable, biodegradable, high porosity, and easy surface modification of nanocellulose-based aerogels. It can be used as a natural adsorbent.

However, because nanocellulose-based aerogels have natural hydroxyl groups on the surface, these hydrophilic groups reduce the adsorption of oily substances and organic solvents. To solve this problem, we usually modify the surface of nanocellulose-based aerogels to increase their hydrophobicity. The hydrophobic function of high-nanofiber-based aerogels is commonly achieved by silylation. Akhlamadi et al. [148] first cross-linked cellulose nanocrystals (CNC) with polyvinyl alcohol (PVA). Then, among them, tetraethyl orthosilicate (TEOS) was added dropwise to prepare silylated PVA/CNC aerogels after freeze-drying. The adsorption experiments demonstrated that the silanized PVA/CNC aerogels showed 69 to 168 g/g adsorption capacity for six oils and eight organic solvents (as shown in Figure 8) and were reusable. After 20 adsorption-extrusion cycles, the aerogel retains more than 92% of its adsorption capacity. The aerogel may be utilized as a reusable adsorbent. Rosli et al. [149] modified cellulose nanocrystals (CNC) from kenaf fibers using methyltrimethoxysilane (MTMS) and combined with γ-irradiated cross-linked gelatin. The γ-irradiated modified cellulose nanocrystals/gelatin aerogels (γ-irradiated CNC-MTMS/gelatin aerogels) were then prepared using the sol-gel method. Its water contact angle (WCA) is 118°. It is shown that the aerogel can absorb crude oil up to 430% of its own weight with good reproducibility. By the eighth cycle, the crude oil uptake had been reduced by only 4%. Zhang et al. [150] used vinyltrimethoxysilane (VTMO) to modify microfibrillated cellulose (MFC). Chopped kapok fiber powder was added to it, followed by the freeze-drying method. High-porosity (99.58%) and hydrophobicity (140.1°) kapok/microfibrillated cellulose aerogels (KCAs) were developed. KCAs have an ultra-high absorption capacity of 104–190.1 g/g, a high retention capacity (of 97%), and a fast oil absorption rate (of 0.74 g/s). Even after 10 uses, it still has a high capacity to soak up oil. Wang et al. [151] used methyltrichlorosilane (MTS) to modify the cellulose nanocrystals. Hydrophobic and lipophilic cellulose nanocrystal (CNC) aerogels (MTS-CNC) were then prepared by gas-phase reactions. MTS-CNC has a high contact angle of 148.5° and a maximum adsorption capacity of 60 g/g for paraffin oil. Qiao et al. [152] placed cellulose nanofibers (CNF) into a polydimethylsiloxane (PDMS) solution for soaking and heat treatment. Cellulose nanofiber (CNF-PDMS) aerogel sheets and blocks with efficient oil/water separation were obtained using directional freeze-drying. CNF-PDMS aerogel sheets can achieve continuous filtration separation of oil and water with a separation efficiency of 99.9%. CNF-PDMS aerogel blocks can separate oil and water mixtures by adsorption and extrusion. It can also be applied to vacuum suction for real-time continuous oil-water separation with a separation efficiency of 99.9%. Figure 9 is a schematic of oil-water separation and CNF-PDMS aerogel sheet and block construction. Mi et al. [153] modified cellulose nanofilaments (CNF) by perfluorododecyltriethoxysilane (PDTS) surface modification combined with the freeze-drying method. They combined organic cellulose nanofilaments (CNF), inorganic silica fibers, and magnetic Fe_3_O_4_ nanoparticles to create a novel ultra-lightweight and superhydrophobic nanocomposite aerogel (M-CNF/silica/Fe_3_O_4_). The aerogel has a high absorption capacity (3420%-5837%) above its own weight, a separation efficiency of 100%, and a WCA of 150°. Liu et al. [154] mixed cellulose nanofibers (CNF) with sodium alginate (SA) and surface-modified by ionic cross-linking and methyltrimethoxysilane (MTMS). Cellulose nanofibers/alginate aerogels (CNF/SA) with a WCA of 144.5° and high porosity (97.85%) were prepared by combining bi-directional freeze-drying. This aerogel possesses a high oil absorption capacity (nearly 88.91 g/g) and is reusable for continuous oil-water separation. It can effectively solve the problem of a marine oil spill. Zhang et al. [155] added vinyltrimethoxysilane (VTMO) and cotton fibers to a cellulose nanofiber (CNF) suspension. After freeze-drying to obtain the wood cotton/cellulose nanofiber aerogel (KNA). Studies have shown that KNA aerogels exhibit high oil absorption (141.9 g/g) as well as outstanding selective adsorption performance. This makes it an ideal sorbent for cleaning oil spills.

There are some other modification methods in the study to make nanocellulose-based aerogels for oil-water separation and the adsorption of organic solvents. To increase the hydrophobicity and the adsorption capacity of nanocellulose (NC) aerogels, Gu et al. [156] prepared NC/NCS/rGO nanocomposite aerogels by introducing reduced graphene oxide (rGO) and nanochitosan (NCS) into NC aerogels by a hydrothermal method combined with freeze-drying. The experiments demonstrated that the adsorption capabilities of the aerogel for acetone, sesame oil, ethyl acetate, mineral oil, thiophene, pump oil, used pump oil, kerosene, and ethanol were 153.22 ± 2.92 g/g, 159.64 ± 1.83 g/g, 149.60 ± 6.26 g/g, 171.85 ± 3.02 g/g, 139.93 ± 3.69 g/g, 132.47 ± 3.45 g/g, 176.82 ± 4.66, 128.70 ± 0.69 g/g, and 120.34 ± 5.57 g/g. NC/NCS/rGO nanocomposite aerogels may effectively remove oil and organic solvents from wastewater. Wang et al. [157] created TOCN aerogels from TEMPO-oxidized cellulose nanofibers (TOCN) by freeze-drying. It was transferred to a tube furnace for carbonization to finally obtain TOCN carbon aerogel. It has been shown that TOCN carbon aerogels have demonstrated superior oil/water selectivity and high absorption. The maximum absorption capacity ranges between 110 g/g and 260 g/g based on the organic solvent density.

In a similar study, Ma et al. [158] added cotton slurry to lithium bromide (LiBr) and stirred it to form a gel after cooling. Then aerogel was prepared by freeze-drying. Finally, the aerogel was put into a tube furnace for carbonization to prepare cellulose carbon aerogel (CCA). The produced aerogel exhibited excellent hydrophobicity (WCA over 135°). It can adsorb up to 55 times its own weight in pump oil. In addition, CCA is reusable, as its ability to absorb oil and organic solvents can be restored to more than 90% of its original capacity after five cycles. Zhou et al. [159] prepared NC/Al_2_O_3_ aerogels by mixing nanocellulose (NC) and nanoalumina (Al_2_O_3_) in deionized water and freeze-drying them. As illustrated in Figure 10, the NC/Al_2_O_3_ aerogel had high adsorption capability for anhydrous ethanol, ethyl acetate, thiophene, cyclohexane, sesame oil, acetone, and dichloromethane in experimental conditions. The adsorption capabilities were 89.91 ± 4.83 g/g, 93.93 ± 3.81 g/g, 108.07 ± 0.37 g/g, 71.13 ± 2.48 g/g, 64.83 ± 2.25 g/g, 85.19 ± 3.87 g/g, and 117.65 ± 5.68 g/g, respectively.

Chhajed et al. [160] prepared hydrophobic aerogels (NLA) by mixing nanocellulose and natural rubber latex (NRL) using the direct mixing method. The WCA of the obtained samples was about 120.5°, with a porous structure that self-assembled. The aerogels had an adsorption capacity of 30–67 g/g in oil/organic solvents and were reused with 85% efficiency for at least 10 cycles. Fan et al. [161] prepared antimicrobial poly (APDMH)-g-ONC (PAC) by grafting 3-(3′-propyl acrylate)-5,5-dimethylglycolide (APDMH) onto oxidized NFC (ONC). PAC and polyethyleneimine (PEI) were chemically cross-linked using 3-glycidoxypropyl trimer (GPTMS). They made PAC-g-PEI aerogels with different kinds of network structures. The schematic diagram of oil-water separation of PAC-g-PEI aerogel is shown in Figure 11. Studies have shown that PAC-g-PEI aerogel achieves greater than 99% oil/water separation efficiency, in excess of 9500 L·m^−2^·h^−1^ of flux. It has a good fatigue resistance of more than fifty compression cycles and a good elasticity of 96.76% height recovery after five compression release cycles at fifty percent strain. Zhang et al. [162] prepared a nanocellulose hybrid aerogel (P-CNS) by UV-induced thiol click reactions. P-CNS aerogel has a high surface area (362.7 m^2^/g) and has adsorption properties on various oils/organic solvents (dichloromethane, soybean oil, pump oil, chloroform, diesel, motor oil, ethanol, acetone, toluene, hexane, gasoline, and octane). The adsorption capacity is 100 to 225 g/g. Zhang et al. [163] successfully synthesized quaternized N-haloamine siloxane monomers. Moreover, blended it with nanocrystalline cellulose (NCC) and chitosan (CS) to prepare efficient oil/water separation and antibacterial aerogels. This aerogel has a high porosity (≥97.66%) and a high separation efficiency (over 99.9%). It has some inhibitory effects on bacteria in oily wastewater.

Table 7 shows the performance of nanocellulose-based aerogels in separating oil/organic solvents and compares them with chitosan-based aerogels.

## 4. Conclusions and Perspectives

Nanocellulose-based aerogel as an emerging porous material. It not only has the three-dimensional structure, high specific surface area, light weight, and high porosity of traditional inorganic aerogel materials. It also has the unique advantages of high biocompatibility, degradability, easy surface modification, good mechanical strength, etc. Nanocellulose-based aerogels with a highly porous three-dimensional network structure exhibit excellent adsorption properties, mechanical strength, and reusability for hazardous substances (dyes, heavy metal ions, petroleum, organic solvents, and antibiotics) in wastewater. As well as the advantages of nanocellulose-based aerogel, such as environmental protection, low cost, a wide source of raw materials, and mechanical durability. Making it significantly better than other adsorbent materials. In addition, 2D nanomaterials such as graphene oxide (GO) and metal organic backbone (MOF) were found to be combined with nanocellulose-based aerogels. Because of the increased porosity and total area, the adsorption characteristics of nanocellulose-based aerogels can be greatly enhanced. Although nanocellulose-based aerogels show excellent performance under harsh conditions and exhibit strong affinity for various hazardous substances in wastewater. However, the application of nanocellulose-based aerogels in wastewater is not without obstacles, and there are still some obstacles to overcome as follows. Here are some key challenges and suggestions for the future:

1. The production of nanocellulose-based aerogels is a complex procedure. Each preparation condition needs to be controlled in order to obtain high-quality products. Moreover, the preparation cost is high, the solvent replacement is time-consuming, and the drying process is complicated. Simpler and more economical methods should be developed to reduce time, cost, energy consumption, and the need for toxic chemicals.

2. Most of the current research on nanocellulose-based aerogels has been carried out in the laboratory. Therefore, the potential of nanocellulose-based aerogels for practical wastewater treatment in industry needs to be evaluated.

3. The regeneration and reuse of nanocellulose-based aerogels is an important issue. Although aerogels can be regenerated and reused by heat treatment, chemical treatment, etc. However, its regeneration effect and utilization times are affected by many factors, such as pollutant type, adsorption amount, regeneration method, etc. Therefore, more economical and efficient methods for regeneration and reuse of aerogels need to be sought to improve the efficiency of aerogel use.

4. The stability and persistence of nanocellulose-based aerogels also need to be further improved. To ensure its stability and adsorption effect during long-term use.

5. The large variety of pollutants in wastewater and the large variation in concentration make it difficult to ensure the adsorption effect of aerogel. In order to address this issue, targeted studies are needed for different types of pollutants.

6. In addition to the variety of pollutants in wastewater, there are also various bacteria present. Therefore, there is a need to further improve the functionality (e.g., antimicrobial properties) of the nanocellulose-based aerogels. So that it has both adsorption capacity to adsorb pollutants and antimicrobial properties to resist bacteria in the wastewater environment.

7. The adsorption mechanism of nanocellulose-based aerogels should be analyzed in more depth. Adequate understanding of the interaction between contaminant molecules and aerogels is needed. This is more conducive to promoting the use of nanocellulose-based aerogels for the removal of new contaminants that may emerge in the water environment in the future.

## Data Availability

Not applicable.

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
