# Peer review of "Preparation of Nanocellulose-Based Aerogel and Its Research Progress in Wastewater Treatment"

_molecules, 2023, doi:10.3390/molecules28083541_

Round 1
Reviewer 1 Report
Comments
This manuscript reviews the fabrication of nanocellulose-based aerogels and discusses the development prospects and future challenges of nanocellulose-based aerogels. And the topic of this article is interesting and timely. However, there are some minor errors that should be addressed before publishing, so I would suggest minor revisions.
1. In part 3.1, the authors inform that "... The adsorption capacity was 270 mg/g for MB and 300 mg/L for MO.” Please check the adsorption capacity unit (mg/L).
2. Authors should pay attention to the abbreviation of professional names, otherwise it is easy to confuse and mislead readers. Such as: “Then aerogel (CA) was prepared by freeze-drying. Finally, the CA was put into a tube furnace for carbonization to prepare cellulose carbon aerogel (CCA). The produced CCA exhibited excellent hydrophobicity (WCA) over 135°).”
3. Manuscript image naming format is not uniformly standardized, authors should follow the MDPI writing format.
4. Some proper nouns such as "tert butanol" (line 223), "tert-butyl alcohol"(line 227, 230), "tert-butanol" (line 233, 263, 351) , "tert" should be italicized.
5. Space is required between the number and unit, such as “418.7mg/g, 291.8mg/g, 128.3mg/g, 230.7mg/g, 227.3mg/g and 454.6mg/g” (line 449), “absorption capacity (nearly 88.91g/g)” (line 552).
6. The manuscript needs careful editing and particular attention to English grammar and spelling.
Author Response
This manuscript reviews the fabrication of nanocellulose-based aerogels and discusses the development prospects and future challenges of nanocellulose-based aerogels. And the topic of this article is interesting and timely. However, there are some minor errors that should be addressed before publishing, so I would suggest minor revisions.
√ Thank you very much for your email. We appreciate the useful comments and suggestions of reviewers, and have revised the manuscript accordingly. Attached please find the revised manuscript for your review. Thank you very much for your consideration for publication in the journal of Molecules.
- In part 3.1, the authors inform that "... The adsorption capacity was 270 mg/g for MB and 300 mg/L for MO.” Please check the adsorption capacity unit (mg/L).
Response: Thank you for your suggestions. It had been changed as followed:
“The adsorption capacity was 270 mg/g for MB and 300 mg/g for MO.”
- Authors should pay attention to the abbreviation of professional names, otherwise it is easy to confuse and mislead readers. Such as: “Then aerogel (CA) was prepared by freeze-drying. Finally, the CA was put into a tube furnace for carbonization to prepare cellulose carbon aerogel (CCA). The produced CCA exhibited excellent hydrophobicity (WCA) over 135°).”
Response: Thank you for your suggestions. It had been changed as followed:
“Then aerogel was prepared by freeze-drying. Finally, the aerogel was put into a tube furnace for carbonization to prepare cellulose carbon aerogel (CCA). The produced aerogel exhibited excellent hydrophobicity (WCA over 135°).”
- Manuscript image naming format is not uniformly standardized, authors should follow the MDPI writing format.
Response: Thank you for your suggestions. The image naming format in the manuscript has been completely revised according to the MDPI format.
- Some proper nouns such as "tert butanol" (line 223), "tert-butyl alcohol"(line 227, 230), "tert-butanol" (line 233, 263, 351) , "tert" should be italicized.
Response: Thank you for your suggestions. It had been changed as followed:
“tert-butanol”
- Space is required between the number and unit, such as “418.7mg/g, 291.8mg/g, 128.3mg/g, 230.7mg/g, 227.3mg/g and 454.6mg/g” (line 449), “absorption capacity (nearly 88.91g/g)” (line 552).
Response: Thank you for your suggestions. It had been changed as followed:
“The adsorption amounts for chloramphenicol, macrolides, quinolones, β-lactams, sulfonamides and tetracyclines were 418.7 mg/g, 291.8 mg/g, 128.3 mg/g, 230.7 mg/g, 227.3 mg/g and 454.6 mg/g, respectively.”
“absorption capacity (nearly 88.91 g/g)”
- The manuscript needs careful editing and particular attention to English grammar and spelling.
Response: Thank you for your suggestions. The syntax and the statements have been modified.

Reviewer 2 Report
This paper presents a comprehensive review of the fabrication of nanocellulose-based aerogels, the development prospects and future challenges of nanocellulose-based aerogels are well-discussed.
However, there are a few areas that could be improved in this paper.
Could the authors enrich the comparation with other natural materials based Aerogels?
The authors may need to summarize the information on applications, such as giving a conclusive table, including comparison of capacity on dyes adsorption/metal ions. I believe that should be helpful for readers to understand the discrepancy more directly.
I think it's the topic original in the field, it includes lots of fundamental knowledge about nanocellulose Aerogels, but there are some format error, for example, the reference style, all the journals name in reference were not in abbreviations, please double check the requirement of< Molecule>.
Author Response
This paper presents a comprehensive review of the fabrication of nanocellulose-based aerogels, the development prospects and future challenges of nanocellulose-based aerogels are well-discussed.
However, there are a few areas that could be improved in this paper.
√ Thank you very much for your email. We appreciate the useful comments and suggestions of reviewers, and have revised the manuscript accordingly. Attached please find the revised manuscript for your review. Thank you very much for your consideration for publication in the journal of Molecules.
Could the authors enrich the comparation with other natural materials based Aerogels?
Response: Thank you for your suggestions. It had been changed as followed:
Table 1. Comparison of the properties of nanocellulose grade aerogels with other natural material-based aerogels
|
Aerogel Category |
Source |
Adjustability |
Mechanical strength |
Renewability |
Production Cost |
Thermal stability |
Mechanical properties |
Water resistance |
Ref.
|
|
Nanocellulose-based aerogel |
Cellulose |
High |
High |
High |
Low |
Low |
High strength High toughness |
Good |
[1] |
|
Starch-based aerogel |
Plant starch |
Low |
Low |
Low |
Low |
Moderate |
Moderate |
Poor |
[2] |
|
Chitosan-based aerogel |
Chitosan |
High |
Moderate |
Moderate |
Moderate |
Moderate |
Moderate |
Good |
[3] |
|
Gelatin-based aerogel |
Animal bones |
Low |
Moderate |
Low |
Moderate |
Moderate |
Moderate |
Poor |
[4] |
The authors may need to summarize the information on applications, such as giving a conclusive table, including comparison of capacity on dyes adsorption/metal ions. I believe that should be helpful for readers to understand the discrepancy more directly.
Response: Thank you for your suggestions. Nanocellulose-based aerogels have been compared with chitosan-based aerogels as shown in Tables 2-4 below.
Table 2. Performance of nanocellulose-based aerogels in dyes removal, compared with chitosan-based aerogels. QCSA: Quaternized chitosan aerogel; SY: Sunset Yellow dyes; HPS: Chitin/chitosan-based aerogel; ZnBDC/CSC: Zn-MOF/Citrate-crosslinked CS.
|
Aerogel name |
Preparation method |
Dyes |
Specific surface area (m2/g) |
Porosity (%) |
Density (mg/cm3) |
Adsorption capacity (mg/g) |
pH |
Number of cycles |
Ref. |
|
Nanocellulose-based aerogels |
|||||||||
|
Cu-BTC/NFC aerogel |
Freeze-drying |
CR |
18.283 |
/ |
/ |
39 |
/ |
/ |
[5] |
|
CGS |
Freeze-drying |
MB |
/ |
98.7 |
19.9 |
608.4 |
7 |
10 |
[6] |
|
SCP |
Freeze-drying |
CR MO |
/ |
/ |
/ |
2007.48 2253.38 |
2-5 |
5 |
[7] |
|
CNF-GnP |
Freeze-drying |
MB CR |
/ |
/ |
/ |
1178.5 585.3 |
/ |
4 |
[8] |
|
Q-CNF |
Freeze-drying |
Bule Red Orange
|
/ |
99 |
17.5 |
230 160 560 |
/ |
20 |
[9] |
|
PCNF |
Freeze-drying |
MB |
368.15 |
/ |
27.2 |
208 |
5 |
5 |
[10] |
|
DADMAC-MBAA modified CNF-Silica aerogels |
Freeze-drying |
MO |
/ |
/ |
/ |
186.7 |
5-7 |
3 |
[11] |
|
ANFs/BC |
Freeze-drying |
MB |
/ |
/ |
/ |
54.45 |
/ |
/ |
[12] |
|
Silica-cellulose aerogel |
Freeze-drying |
MB MO |
350 |
93 |
107 |
270 300 |
/ |
/ |
[13] |
|
Cellulose nanofibril-based aerogel |
Freeze-drying |
MG |
193 |
/ |
/ |
212.7 |
/ |
4 |
[14] |
|
CO2-responsive cellulose nanofibril aerogel |
Freeze-drying |
MB NGB MO |
17.97 |
/ |
21.7 |
598.8 621.1 892.9 |
/ |
20 |
[15] |
|
Chitosan-based aerogels |
|||||||||
|
QCSA |
Freeze-drying |
CR MO SY |
/ |
/ |
60 |
1259.6 607.5 550.2 |
7 |
5 |
[16] |
|
HPS |
Freeze-drying |
CR |
123.92 |
98.16 |
/ |
2074 |
6 |
7 |
[17] |
|
Fc-CS |
Freeze-drying |
MB |
5 |
/ |
/ |
156.3 |
5-6 |
5 |
[18] |
|
ZnBDC/CSC |
Freeze-drying |
MO |
16.5 |
/ |
/ |
202 |
5 |
5 |
[19] |
Table 3. Performance of nanocellulose-based aerogels in metal ion removal, compared with chitosan-based aerogels. PCA: Polyethyleneimine (PEI) functionalized chitosan (CS) aerogel; CSTU: Thiourea-chitosan (CSTU) aerogel; WP-CSA: Waste paper/chitosan aerogel; E-CS aerogel: Enhanced chitosan aerogel; CS-MMT: Chitosan-montmorillonite composite aerogel.
|
Aerogel name |
Preparation method |
Metal ions |
Specific surface area (m2/g) |
Porosity (%) |
Density (mg/cm3) |
Adsorption capacity (mg/g) |
pH |
Number of cycles |
Ref. |
|
Nanocellulose-based aerogels |
|||||||||
|
APTMs modified TO-NFC |
Freeze-drying |
Cu (â…¡) Cd (â…¡) Hg (â…¡) |
129.32 |
99.14 |
/ |
99 124.5 242.1 |
3-7 |
/ |
[20] |
|
TOCNF-TMPTAP-APAM |
Freeze-drying |
Cu (â…¡) |
/ |
99.1 |
14.4 |
240.00 |
6 |
10 |
[21] |
|
MOF@CA |
Freeze-drying |
Pb (â…¡) Cu (â…¡) |
/ |
99.4-99.5 |
9.8-11.2 |
123.00 70.53 |
/ |
5 |
[22] |
|
NFC/PEI hybrid aerogels |
Freeze-drying |
Pb (â…¡) Cu (â…¡) |
42.5 |
/ |
/ |
357.44 175.44 |
2-5 |
3 |
[23] |
|
CPA |
Freeze-drying |
Cr (â…¥) |
36.77 |
/ |
/ |
229.10 |
2 |
5 |
[24] |
|
PEI@CNF aerogels |
Freeze-drying |
Cu (â…¡) |
11.48 |
/ |
/ |
135.10 |
3-6 |
3 |
[25] |
|
nanocellulose-Fe3O4 hybrid aerogel |
Freeze-drying |
Cr (â…¡) Pb (â…¡) Cu (â…¡) |
/ |
5 |
/ |
2.20 1.25 0.40 |
/ |
/ |
[26] |
|
UiO-66-NH2@CA |
Freeze-drying |
Pb (â…¡) |
/ |
/ |
/ |
89.40 |
/ |
5 |
[27] |
|
CNFs aerogel |
Freeze-drying |
U (â…¥) |
188 |
/ |
/ |
440.60 |
5 |
|
[28] |
|
CGP |
Freeze-drying |
Cr (â…¥) |
/ |
/ |
/ |
386.40 |
2 |
5 |
[29] |
|
Chitosan-based aerogels |
|||||||||
|
PCA |
Freeze-drying |
Cr (â…¥) |
/ |
/ |
/ |
445.29 |
3 |
10 |
[30] |
|
CSTU |
Freeze-drying |
Ag (â… ) Pb (â…¡) |
416.64-447.26 |
/ |
2.1-10.3 |
1.11mmol/g 0.48mmol/g |
6 |
5 |
[31] |
|
WP-CSA |
Freeze-drying |
Cu (â…¡) |
/ |
/ |
106 |
156.3 |
2.3-5.5 |
/ |
[32] |
|
E-CS aerogel |
Freeze-drying |
Cu (â…¡) Pb (â…¡) Cd (â…¡) |
/ |
97.38 |
38.3 |
108.14 143.73 84.62 |
5 |
3 |
[33] |
|
CS-MMT |
Freeze-drying |
Cu (â…¡) |
14.133 |
/ |
/ |
86.95 |
6 |
7 |
[34] |
Table 4. Performance of nanocellulose-based aerogels in antibiotics removal, compared with chitosan-based aerogels. GO/CNF: GO/ nanofibrillated cellulose aerogel; PGO-CS: Porous graphene oxide–chitosan aerogel; CMC: Carboxymethyl chitosan aerogel; CMC-MT: Na-montmorillonite (Na-Mt) with carboxymethyl chitosan (CMC).
|
Aerogel name |
Preparation method |
Antibiotics |
Specific surface area (m2/g) |
Porosity (%) |
Adsorption capacity (mg/g) |
pH |
Number of cycles |
Ref. |
|
|
Nanocellulose-based aerogels |
|||||||||
|
CNF/GO |
Freeze-drying |
Loramphenicol Macrolides Quinolones β-lactams sulfonamides tetracyclines |
97.5 |
/ |
418.7 291.8 128.3 230.7 227.3 454.6 |
2.0 |
10 |
[35] |
|
|
GO-CNF |
Freeze-drying |
DXC CTC OTC TC |
89.9 |
/ |
469.7 396.5 386.5 343.8 |
/ |
5 |
[36] |
|
|
GO/CNF |
Freeze-drying |
TC |
35 |
/ |
47.3 |
/ |
3 |
[37] |
|
|
BCCA |
Freeze-drying |
CAP NOR SMX |
1505 |
/ |
525 1926 1264 |
5 |
5 |
[38] |
|
|
ZIF-67 / PANI / RCA aerogel |
Freeze-drying |
TC
|
/ |
/ |
409.55 |
7.0 |
6 |
[39] |
|
|
ZCCA |
Freeze-drying |
ENR |
756.45 |
95 |
172.09 |
2.0-6.0 |
6 |
[40] |
|
|
PVA-assisted CNCs/SiO2 aerogel |
Freeze-drying |
CIP
|
/ |
/ |
163.34 |
4.0 |
/ |
[41] |
|
|
Chitosan-based aerogels |
|||||||||
|
PGO-CS |
Freeze-drying |
TC |
345 |
/ |
1470 |
9.0-10.0 |
4 |
[42] |
|
|
CMC |
Freeze-drying |
TC |
0.73 |
/ |
332.23 |
3.0-4.0 |
/ |
[43] |
|
|
CMC-Mt |
Freeze-drying |
CTC |
119.526 |
/ |
48.71 |
4.0-7.0 |
/ |
[44] |
|
Table 5. Performance of nanocellulose-based aerogels compared with chitosan-based aerogels for oil/organic solvents separation. PNI-Si@CCNT/CA: Vinyltrimethoxysilane and thermally responsive poly(N-isopropylacrylamide) (PNIPAAm) were grafted onto the surface of carboxylated carbon nanotube/chitosan aerogel backbone to obtain aerogels; CA/CS/CMC: Citric acid/Chitosan/Carboxymethyl cellulose aerogel; DMF: Dimethylformamide; THF: Tetrahydrofuran; DMSO: Dimethyl sulfoxide; DCM: Dichloromethane.
|
Aerogel name |
Preparation method |
Oil/Organic solvents |
Specific surface area (m2/g) |
Porosity (%) |
Density (mg/cm3) |
Adsorption capacity (g/g) |
Water contact angle (°) |
Number of cycles |
Ref. |
|
Nanocellulose-based aerogels |
|||||||||
|
silylated PVA/CNC aerogels |
Freeze-drying |
Chloroform Dodecane Acetone Ethanol DMF 2-Propanol Etyl acetate Hexane Toluene Xylene Olive oil Cooked oil Sesame oil Motor oil Crude oil Gasoline |
76 |
98.42 |
17 |
69-168 |
154.93 |
20 |
[45] |
|
γ-irradiated CNC-MTMS/gelatin aerogels |
Freeze-drying |
Chloroform Crude oil |
/ |
/ |
85 |
/ (It can absorb 430% of its own weight) |
118 |
8 |
[46] |
|
KCAs |
Freeze-drying |
Vegetable Oil Motor oil Gasoline Vacuum pump oil Trichloromethane Ethanol DMF |
/ |
99.58 |
5.1 |
104-190.2 |
140.1 |
10 |
[47] |
|
MTS-CNC |
Freeze-drying |
liquid paraffin oil |
282 |
/ |
/ |
60 |
148.5 |
5 |
[48] |
|
CNF-PDMS |
Freeze-drying |
Dim ethylb enzene Ethyl acetate Ethan ol n-Hexane n-Decane n-Dodecane n-Hexadecane Methylcyclohexane Dichloroethane Toluene Dimethylformamide Petroleum ether Tetrahydrofuran Petroleum |
/ |
98.4 |
22.7 |
24-48 |
163.5 |
20 |
[49] |
|
M-CNF/silica/Fe3O4) |
Freeze-drying |
DMF DMSO Octane Gasoline Dioxane Toluene Hexane Chloroform |
82.6 |
/ |
22.3 |
34-58 |
150 |
10 |
[50] |
|
CNF/SA |
Freeze-drying |
Flax seed oil Pump oil Used pump oil Olive oil Silicane oil Toluene Acetone Ethanol Hexane Ethylene glycol DMF DMSO |
149.64 |
97.85 |
24.2 |
41.16-88.91 |
144.5 |
20 |
[51] |
|
KNA |
Freeze-drying |
vegetable oil |
/ |
99.5-99.6 |
4.9-6.0 |
141.9 |
147.6 |
/ |
[52] |
|
NC/NCS/rGO nanocomposite aerogel |
Freeze-drying |
Acetone sesame oil ethyl acetate mineral oil thiophene pump oil used pump oil kerosene ethanol |
/ |
99.18 |
/ |
153.22 159.64 149.60 171.85 139.93 132.47 176.82 128.70 120.34
|
115.26 |
/ |
[53] |
|
TOCN carbon aerogel |
Freeze-drying and high-temperature carbonization |
Gasoline Diesel oil Pump oil Motor oil Sesame oil Chloroform Acetaldehyde Ethanol Toluene Octadecylene Cyclohexane Heptane n-Hexane Acetone Methanol Lactic acid Styrene THF DMF |
249.91 |
99.5 |
8.8 |
110-260 |
139.6 |
5 |
[54] |
|
CCA |
Freeze-drying and high-temperature carbonization |
Soybean oil Pump oil Acetone Ethylene glycol Methanol DMF Hexane Ethanol |
79.2 |
98.9-99.2 |
16-23 |
22-55 |
>135 |
5 |
[55] |
|
NC/Al2O3 aerogel |
Freeze-drying |
anhydrous ethanol ethyl acetate thiophene cyclohexane sesame oil acetone dichlormethae |
124 |
99.09 |
5.1 |
89.91 93.93 108.07 71.13 64.83 85.19 117.65 |
/ |
/ |
[56] |
|
NLA |
Freeze-drying |
Crude oil Rexid oil Silicon oil Vacuum pump oil Red oil Hexane Xylene DMF THF DCM Chloroform |
/ |
98 |
22 |
30-67 |
120.5 |
10 |
[57] |
|
PAC-g-PEI |
Freeze-drying |
n-hexane toluene edible oil silicone oil |
/ |
94 |
67 |
/ (Separation efficiency over 99%) |
/ (Oil contact angle is 130.3°-135°) |
50 |
[58] |
|
P-CNS |
Freeze-drying |
dichloromethae soybean oil pump oil chloroform diesel motor oil ethanol acetone toluene hexane gasoline octane |
362.7 |
98.9-99.4 |
8.4-12.9 |
100-225 |
133.6-168.4 |
50 |
[59] |
|
NCC/CS aerogel |
Freeze-drying |
Methylbenzene Petroleum ether n-hexane edible oil silicone oil and dodecane |
/ |
97.66 |
40.82 |
/ (Separation efficiency over 99%) |
/ (Oil contact angle is 109°-141.1°) |
50 |
[60] |
|
Chitosan-based aerogels |
|||||||||
|
PNI-Si@CCNT/CA |
Freeze-drying |
N-hexane Toluene Trichloromethane Petroleum ether Peanut oil Soybean oil Sunflower oil Olive oil
|
2.81 |
/ |
0.0051 |
23.8 35.3 53.0 42.1 41.0 35.2 32.5 40.8 |
/ (Oil contact angle is 134°) |
9 |
[61] |
|
CsA |
Freeze-drying |
Crude oil Diesel |
28.3 |
97.98 |
28.3 |
41.07 31.07 |
/ |
3.0-4.0 |
[62] |
|
(CA/CS/CMC) |
Freeze-drying |
Chloroform Toluene Acetone Methanol Ethanol |
/ |
96 |
8.3-63.6 |
27-44 |
/ |
4.0-7.0 |
[63] |
I think it's the topic original in the field, it includes lots of fundamental knowledge about nanocellulose Aerogels, but there are some format error, for example, the reference style, all the journals name in reference were not in abbreviations, please double check the requirement of< Molecule>.
Response: Thank you for your suggestions. The references have been inserted in the endnote format of the Molecules.

Reviewer 3 Report
Attached..

Author Response
Comments to authors.
The manuscript molecules-2328625 entitled, “Preparation of Nanocellulose-based Aerogel and its Research Progress in Wastewater Treatment " reports a utilization of nanocellulose for aerogel formation, and its application in the fields of pollutant removal from waste water. Though the intention of the authors is commendable, there are some problems in the manuscript that need to be revised. For a better understanding, arrange all the literatures the tabular format with relevant references. Some revision details are given in comments below.
√ Thank you very much for your email. We appreciate the useful comments and suggestions of reviewers, and have revised the manuscript accordingly. Attached please find the revised manuscript for your review. Thank you very much for your consideration for publication in the journal of Molecules.
In abstract, Line 23: Remove ‘Secondly’ from the sentence and user ‘Furthermore’ or use ‘Firstly’
in previous sentence.
Response: Thank you for your suggestions. It had been changed as followed:
“Furthermore, the research progress of the application of nanocellulose-based aerogels in the adsorption of dyes, heavy metal ions, antibiotics, organic solvents and oil-water separation is reviewed.”
Introduction: Line 31-32-Aviod to repeat the word ‘rapid’ in the same sentence.
Response: Thank you for your suggestions. It had been changed as followed:
“However, the rapid growth of the world population, the construction of modern cities and the development of industrialization have led to the pollution of a large amount of water quality environment.”
Scheme 1. Should also be appears in the main text
Response: Thank you for your suggestions. Scheme 1 has been marked in the manuscript.
Line 73: Please do not use personal pronoun like “We make” in the sentence.
Response: Thank you for your suggestions. It had been changed as followed:
“Lastly, recommendations for future research, which explore its challenges and shortcomings and provide strategies for future study.”
Preparation of nanocellulose-based aerogel
Line: 78-79: Please rephrase this sentence as the word ‘renewable of renewable’ hard to understand
Response: Thank you for your suggestions. It had been changed as followed:
“It not only retains the same qualities as traditional aerogel, but it also has the attributes of renewable, green, non-toxic, low-cost, and biodegradable nanocellulose.”
Preparation of nanocellulose
This section must be subdivided in to, Chemical methods including well known sulphuric acid and other chemicals (Please note TEMPO is the chemical oxidation of nanocellulose and can be used as post or pretreatments to introduce acid functional group to the cellulose backbone), mechanical method, physico-chemical and enzymatic/bacterial etc. methods. Briefly comments on advantages and disadvantages associated with these methods. Categories, aerogel based on cellulose nanocrystals (CNC), or particles, cellulose nanofilaments/fibers (CNF) and bacterial cellulose(BC) etc. separately. Arrange all these in tubular format indicating its sources, methods employed, properties, and application in the removing of the particular pollutant and comparison with existing commercial one. This will give in depth knowledge and understanding of the readers.
Response: Thank you for your suggestions. It had been changed as followed:
Table 1. Classification of nanocellulose.
|
Category |
Source |
Preparation method |
Advantages |
Disadvantages |
Ref. |
|
Cellulose nanocrystals (CNC) |
Cellulose |
Acid hydrolysis High shear mechanical stripping |
High specific surface area High mechanical properties Biodegradable |
Higher production costs Easy to gather |
[1] |
|
Cellulose nanofilaments (CNF) |
Cellulose |
Acid hydrolysis High shear mechanical stripping Biological preparation Chemical oxidative stripping |
High specific surface area High mechanical properties Biodegradable Can be prepared into a variety of forms |
Higher production costs Easy to gather |
[2] |
|
Bacterial cellulose (BC) |
Natural cellulose material synthesized by microbial growth |
Extraction from cultures by chemical and physical methods |
Biodegradable Can be prepared in a variety of forms |
Higher production cost Poorer mechanical properties |
[3] |
Table 2. Classification of nanocellulose preparation methods.
|
Preparation method |
Advantages |
Disadvantages |
Ref. |
|
Mechanical preparation |
Simple operation No chemical reagents required |
Limited production capacity Requiring high energy consumption equipment |
[4] |
|
Chemical preparation |
Can precisely control the structure and morphology of the product |
The use of chemical reagents is harmful to human body High environmental impact |
[5] |
|
Bio-enzyme preparation |
The production process is environmentally friendly The prepared nanocellulose has a uniform structure High cellulose decomposition rate |
Enzyme preparations are expensive Take a long time to prepare |
[6,7] |
|
Acid hydrolysis preparation |
Low cost Simple and easy to use stable and controllable Quality of finished products |
Production environment with acidic wastewater discharge easy to produce by-products Need to treat wastewater and waste acid |
[8,9] |
|
TEMPO oxidation method preparation |
The production process is environmentally friendly The prepared nanocellulose has a uniform structure High cellulose decomposition rate |
TEMPO reagents are expensive Take a long time to prepare |
[10] |
Line 125-126: Liuet al. [42] obtained lignocellulosic nanofibers (LCNFs) with a network structure of 15-30 nm…. Why authors also describing the lignocellulose nanofibers? Be consistent with the pure nanocellulose only.
Response: Thank you for your suggestions. This section has been removed.
Line 291: Avoid to use personal pronoun like ‘we urgently’……………………in the sentence.
Response: Thank you for your suggestions. It had been changed as followed:
“Therefore, there is a global need to solve the problem of dye pollution.”
In all other subsequent section, applications of aerogels for particular pollutant system must be tabulated with detail information along with relevant references.
Response: Thank you for your suggestions. It had been changed as followed:
|
Aerogel name |
Preparation method |
Dyes |
Specific surface area (m2/g) |
Porosity (%) |
Density (mg/cm3) |
Adsorption capacity (mg/g) |
pH |
Number of cycles |
Ref. |
|
Nanocellulose-based aerogels |
|||||||||
|
Cu-BTC/NFC aerogel |
Freeze-drying |
CR |
18.283 |
/ |
/ |
39 |
/ |
/ |
[11] |
|
CGS |
Freeze-drying |
MB |
/ |
98.7 |
19.9 |
608.4 |
7 |
10 |
[12] |
|
SCP |
Freeze-drying |
CR MO |
/ |
/ |
/ |
2007.48 2253.38 |
2-5 |
5 |
[13] |
|
CNF-GnP |
Freeze-drying |
MB CR |
/ |
/ |
/ |
1178.5 585.3 |
/ |
4 |
[14] |
|
Q-CNF |
Freeze-drying |
Bule Red Orange
|
/ |
99 |
17.5 |
230 160 560 |
/ |
20 |
[15] |
|
PCNF |
Freeze-drying |
MB |
368.15 |
/ |
27.2 |
208 |
5 |
5 |
[16] |
|
DADMAC-MBAA modified CNF-Silica aerogels |
Freeze-drying |
MO |
/ |
/ |
/ |
186.7 |
5-7 |
3 |
[17] |
|
ANFs/BC |
Freeze-drying |
MB |
/ |
/ |
/ |
54.45 |
/ |
/ |
[18] |
|
Silica-cellulose aerogel |
Freeze-drying |
MB MO |
350 |
93 |
107 |
270 300 |
/ |
/ |
[19] |
|
Cellulose nanofibril-based aerogel |
Freeze-drying |
MG |
193 |
/ |
/ |
212.7 |
/ |
4 |
[20] |
|
CO2-responsive cellulose nanofibril aerogel |
Freeze-drying |
MB NGB MO |
17.97 |
/ |
21.7 |
598.8 621.1 892.9 |
/ |
20 |
[21] |
|
Chitosan-based aerogels |
|||||||||
|
QCSA |
Freeze-drying |
CR MO SY |
/ |
/ |
60 |
1259.6 607.5 550.2 |
7 |
5 |
[22] |
|
HPS |
Freeze-drying |
CR |
123.92 |
98.16 |
/ |
2074 |
6 |
7 |
[23] |
|
Fc-CS |
Freeze-drying |
MB |
5 |
/ |
/ |
156.3 |
5-6 |
5 |
[24] |
|
ZnBDC/CSC |
Freeze-drying |
MO |
16.5 |
/ |
/ |
202 |
5 |
5 |
[25] |
Table 3. Performance of nanocellulose-based aerogels in dyes removal, compared with chitosan-based aerogels. QCSA: Quaternized chitosan aerogel; SY: Sunset Yellow dyes; HPS: Chitin/chitosan-based aerogel; ZnBDC/CSC: Zn-MOF/Citrate-crosslinked CS.
Table 4. Performance of nanocellulose-based aerogels in metal ion removal, compared with chitosan-based aerogels. PCA: Polyethyleneimine (PEI) functionalized chitosan (CS) aerogel; CSTU: Thiourea-chitosan (CSTU) aerogel; WP-CSA: Waste paper/chitosan aerogel; E-CS aerogel: Enhanced chitosan aerogel; CS-MMT: Chitosan-montmorillonite composite aerogel.
|
Aerogel name |
Preparation method |
Metal ions |
Specific surface area (m2/g) |
Porosity (%) |
Density (mg/cm3) |
Adsorption capacity (mg/g) |
pH |
Number of cycles |
Ref. |
|
Nanocellulose-based aerogels |
|||||||||
|
APTMs modified TO-NFC |
Freeze-drying |
Cu (â…¡) Cd (â…¡) Hg (â…¡) |
129.32 |
99.14 |
/ |
99 124.5 242.1 |
3-7 |
/ |
[26] |
|
TOCNF-TMPTAP-APAM |
Freeze-drying |
Cu (â…¡) |
/ |
99.1 |
14.4 |
240.00 |
6 |
10 |
[27] |
|
MOF@CA |
Freeze-drying |
Pb (â…¡) Cu (â…¡) |
/ |
99.4-99.5 |
9.8-11.2 |
123.00 70.53 |
/ |
5 |
[28] |
|
NFC/PEI hybrid aerogels |
Freeze-drying |
Pb (â…¡) Cu (â…¡) |
42.5 |
/ |
/ |
357.44 175.44 |
2-5 |
3 |
[29] |
|
CPA |
Freeze-drying |
Cr (â…¥) |
36.77 |
/ |
/ |
229.10 |
2 |
5 |
[30] |
|
PEI@CNF aerogels |
Freeze-drying |
Cu (â…¡) |
11.48 |
/ |
/ |
135.10 |
3-6 |
3 |
[31] |
|
nanocellulose-Fe3O4 hybrid aerogel |
Freeze-drying |
Cr (â…¡) Pb (â…¡) Cu (â…¡) |
/ |
5 |
/ |
2.20 1.25 0.40 |
/ |
/ |
[32] |
|
UiO-66-NH2@CA |
Freeze-drying |
Pb (â…¡) |
/ |
/ |
/ |
89.40 |
/ |
5 |
[33] |
|
CNFs aerogel |
Freeze-drying |
U (â…¥) |
188 |
/ |
/ |
440.60 |
5 |
|
[34] |
|
CGP |
Freeze-drying |
Cr (â…¥) |
/ |
/ |
/ |
386.40 |
2 |
5 |
[35] |
|
Chitosan-based aerogels |
|||||||||
|
PCA |
Freeze-drying |
Cr (â…¥) |
/ |
/ |
/ |
445.29 |
3 |
10 |
[36] |
|
CSTU |
Freeze-drying |
Ag (â… ) Pb (â…¡) |
416.64-447.26 |
/ |
2.1-10.3 |
1.11mmol/g 0.48mmol/g |
6 |
5 |
[37] |
|
WP-CSA |
Freeze-drying |
Cu (â…¡) |
/ |
/ |
106 |
156.3 |
2.3-5.5 |
/ |
[38] |
|
E-CS aerogel |
Freeze-drying |
Cu (â…¡) Pb (â…¡) Cd (â…¡) |
/ |
97.38 |
38.3 |
108.14 143.73 84.62 |
5 |
3 |
[39] |
|
CS-MMT |
Freeze-drying |
Cu (â…¡) |
14.133 |
/ |
/ |
86.95 |
6 |
7 |
[40] |
Table 5. Performance of nanocellulose-based aerogels in antibiotics removal, compared with chitosan-based aerogels. GO/CNF: GO/ nanofibrillated cellulose aerogel; PGO-CS: Porous graphene oxide–chitosan aerogel; CMC: Carboxymethyl chitosan aerogel; CMC-MT: Na-montmorillonite (Na-Mt) with carboxymethyl chitosan (CMC).
|
Aerogel name |
Preparation method |
Antibiotics |
Specific surface area (m2/g) |
Porosity (%) |
Adsorption capacity (mg/g) |
pH |
Number of cycles |
Ref. |
|
|
Nanocellulose-based aerogels |
|||||||||
|
CNF/GO |
Freeze-drying |
Loramphenicol Macrolides Quinolones β-lactams sulfonamides tetracyclines |
97.5 |
/ |
418.7 291.8 128.3 230.7 227.3 454.6 |
2.0 |
10 |
[41] |
|
|
GO-CNF |
Freeze-drying |
DXC CTC OTC TC |
89.9 |
/ |
469.7 396.5 386.5 343.8 |
/ |
5 |
[42] |
|
|
GO/CNF |
Freeze-drying |
TC |
35 |
/ |
47.3 |
/ |
3 |
[43] |
|
|
BCCA |
Freeze-drying |
CAP NOR SMX |
1505 |
/ |
525 1926 1264 |
5 |
5 |
[44] |
|
|
ZIF-67 / PANI / RCA aerogel |
Freeze-drying |
TC
|
/ |
/ |
409.55 |
7.0 |
6 |
[45] |
|
|
ZCCA |
Freeze-drying |
ENR |
756.45 |
95 |
172.09 |
2.0-6.0 |
6 |
[46] |
|
|
PVA-assisted CNCs/SiO2 aerogel |
Freeze-drying |
CIP
|
/ |
/ |
163.34 |
4.0 |
/ |
[47] |
|
|
Chitosan-based aerogels |
|||||||||
|
PGO-CS |
Freeze-drying |
TC |
345 |
/ |
1470 |
9.0-10.0 |
4 |
[48] |
|
|
CMC |
Freeze-drying |
TC |
0.73 |
/ |
332.23 |
3.0-4.0 |
/ |
[49] |
|
|
CMC-Mt |
Freeze-drying |
CTC |
119.526 |
/ |
48.71 |
4.0-7.0 |
/ |
[50] |
|
Table 6. Performance of nanocellulose-based aerogels compared with chitosan-based aerogels for oil/organic solvents separation. PNI-Si@CCNT/CA: Vinyltrimethoxysilane and thermally responsive poly(N-isopropylacrylamide) (PNIPAAm) were grafted onto the surface of carboxylated carbon nanotube/chitosan aerogel backbone to obtain aerogels; CA/CS/CMC: Citric acid/Chitosan/Carboxymethyl cellulose aerogelï¼›DMF: Dimethylformamide; THF: Tetrahydrofuran; DMSO: Dimethyl sulfoxide; DCM: Dichloromethane.
|
Aerogel name |
Preparation method |
Oil/Organic solvents |
Specific surface area (m2/g) |
Porosity (%) |
Density (mg/cm3) |
Adsorption capacity (g/g) |
Water contact angle (°) |
Number of cycles |
Ref. |
|
Nanocellulose-based aerogels |
|||||||||
|
silylated PVA/CNC aerogels |
Freeze-drying |
Chloroform Dodecane Acetone Ethanol DMF 2-Propanol Etyl acetate Hexane Toluene Xylene Olive oil Cooked oil Sesame oil Motor oil Crude oil Gasoline |
76 |
98.42 |
17 |
69-168 |
154.93 |
20 |
[51] |
|
γ-irradiated CNC-MTMS/gelatin aerogels |
Freeze-drying |
Chloroform Crude oil |
/ |
/ |
85 |
/ (It can absorb 430% of its own weight) |
118 |
8 |
[52] |
|
KCAs |
Freeze-drying |
Vegetable Oil Motor oil Gasoline Vacuum pump oil Trichloromethane Ethanol DMF |
/ |
99.58 |
5.1 |
104-190.2 |
140.1 |
10 |
[53] |
|
MTS-CNC |
Freeze-drying |
liquid paraffin oil |
282 |
/ |
/ |
60 |
148.5 |
5 |
[54] |
|
CNF-PDMS |
Freeze-drying |
Dim ethylb enzene Ethyl acetate Ethan ol n-Hexane n-Decane n-Dodecane n-Hexadecane Methylcyclohexane Dichloroethane Toluene Dimethylformamide Petroleum ether Tetrahydrofuran Petroleum |
/ |
98.4 |
22.7 |
24-48 |
163.5 |
20 |
[55] |
|
M-CNF/silica/Fe3O4) |
Freeze-drying |
DMF DMSO Octane Gasoline Dioxane Toluene Hexane Chloroform |
82.6 |
/ |
22.3 |
34-58 |
150 |
10 |
[56] |
|
CNF/SA |
Freeze-drying |
Flax seed oil Pump oil Used pump oil Olive oil Silicane oil Toluene Acetone Ethanol Hexane Ethylene glycol DMF DMSO |
149.64 |
97.85 |
24.2 |
41.16-88.91 |
144.5 |
20 |
[57] |
|
KNA |
Freeze-drying |
vegetable oil |
/ |
99.5-99.6 |
4.9-6.0 |
141.9 |
147.6 |
/ |
[58] |
|
NC/NCS/rGO nanocomposite aerogel |
Freeze-drying |
Acetone sesame oil ethyl acetate mineral oil thiophene pump oil used pump oil kerosene ethanol |
/ |
99.18 |
/ |
153.22 159.64 149.60 171.85 139.93 132.47 176.82 128.70 120.34
|
115.26 |
/ |
[59] |
|
TOCN carbon aerogel |
Freeze-drying and high-temperature carbonization |
Gasoline Diesel oil Pump oil Motor oil Sesame oil Chloroform Acetaldehyde Ethanol Toluene Octadecylene Cyclohexane Heptane n-Hexane Acetone Methanol Lactic acid Styrene THF DMF |
249.91 |
99.5 |
8.8 |
110-260 |
139.6 |
5 |
[60] |
|
CCA |
Freeze-drying and high-temperature carbonization |
Soybean oil Pump oil Acetone Ethylene glycol Methanol DMF Hexane Ethanol |
79.2 |
98.9-99.2 |
16-23 |
22-55 |
>135 |
5 |
[61] |
|
NC/Al2O3 aerogel |
Freeze-drying |
anhydrous ethanol ethyl acetate thiophene cyclohexane sesame oil acetone dichlormethae |
124 |
99.09 |
5.1 |
89.91 93.93 108.07 71.13 64.83 85.19 117.65 |
/ |
/ |
[62] |
|
NLA |
Freeze-drying |
Crude oil Rexid oil Silicon oil Vacuum pump oil Red oil Hexane Xylene DMF THF DCM Chloroform |
/ |
98 |
22 |
30-67 |
120.5 |
10 |
[63] |
|
PAC-g-PEI |
Freeze-drying |
n-hexane toluene edible oil silicone oil |
/ |
94 |
67 |
/ (Separation efficiency over 99%) |
/ (Oil contact angle is 130.3°-135°) |
50 |
[64] |
|
P-CNS |
Freeze-drying |
dichloromethae soybean oil pump oil chloroform diesel motor oil ethanol acetone toluene hexane gasoline octane |
362.7 |
98.9-99.4 |
8.4-12.9 |
100-225 |
133.6-168.4 |
50 |
[65] |
|
NCC/CS aerogel |
Freeze-drying |
Methylbenzene Petroleum ether n-hexane edible oil silicone oil and dodecane |
/ |
97.66 |
40.82 |
/ (Separation efficiency over 99%) |
/ (Oil contact angle is 109°-141.1°) |
50 |
[66] |
|
Chitosan-based aerogels |
|||||||||
|
PNI-Si@CCNT/CA |
Freeze-drying |
N-hexane Toluene Trichloromethane Petroleum ether Peanut oil Soybean oil Sunflower oil Olive oil
|
2.81 |
/ |
0.0051 |
23.8 35.3 53.0 42.1 41.0 35.2 32.5 40.8 |
/ (Oil contact angle is 134°) |
9 |
[67] |
|
CsA |
Freeze-drying |
Crude oil Diesel |
28.3 |
97.98 |
28.3 |
41.07 31.07 |
/ |
3.0-4.0 |
[68] |
|
(CA/CS/CMC) |
Freeze-drying |
Chloroform Toluene Acetone Methanol Ethanol |
/ |
96 |
8.3-63.6 |
27-44 |
/ |
4.0-7.0 |
[69] |
Moreover, improvement the English language throughout the manuscript and needed.
Response: Thank you for your suggestions. English grammar and word spelling have been checked and perfected.

Round 2
Reviewer 3 Report
Most of my concerns have been satisfactorily addressed by the authors. In particular, all pertinent results have been tabulated so that the reader can compare and comprehend them easily.